# Predictors of Post-Traumatic Stress Symptoms after musculoskeletal trauma

Ferozkhan Jadhakhan[1,2]*, Deborah Falla[3]*, Alexander Dallaway[2,4]

**1** School of Life and Health Sciences, Birmingham City University, City South Campus, Birmingham, United Kingdom, **2** School of Health and Wellbeing, Faculty of Education, Health and Wellbeing, University of Wolverhampton, Wolverhampton, United Kingdom, **3** Centre of Precision Rehabilitation for Spinal Pain (CPR Spine), School of Sport, Exercise and Rehabilitation Sciences, College of Life and Environmental Sciences, University of Birmingham, Birmingham, United Kingdom, **4** Warwickshire Institute for the Study of Diabetes, Endocrinology and Metabolism (WISDEM), University Hospitals Coventry and Warwickshire NHS Trust, Coventry, United Kingdom

* ferozkhan.jadhakhan@bcu.ac.uk; d.falla@bham.ac.uk

## Abstract

### Background

Post-traumatic stress symptoms (PTSS) are common following musculoskeletal trauma and are associated with poorer recovery, disability, and reduced quality of life. Although psychosocial and injury-related factors have been linked to PTSS, limited research has examined longitudinal predictors or the relationship between injury severity and PTSS progression in people hospitalised for musculoskeletal trauma.

### Objectives

To identify predictors of PTSS at three and six months following musculoskeletal trauma and hospitalisation, and to examine the relationship between injury severity and PTSS severity over time.

### Methods

This secondary analysis of a prospective cohort study included 125 adults admitted to a UK major trauma centre with acute musculoskeletal injuries. PTSS were assessed using the Impact of Event Scale–Revised (IES-R) at baseline, three, and six months, with scores ≥22 indicating elevated PTSS. Candidate predictors included socio-demographic, clinical, and trauma-related variables. Multivariate logistic regression identified predictors at three and six months. Multicollinearity was assessed using variance inflation factors and principal component analysis. Injury severity (mild, moderate, major) was examined using Kruskal-Wallis and Mann-Whitney U tests. Model discrimination and calibration were evaluated using AUC and Hosmer-Lemeshow tests.

**Data availability statement:** All relevant data are within the paper and its Supporting Information files.

**Funding:** The author(s) received no specific funding for this work.

**Competing interests:** The authors have declared that no competing interests exist.

## Results

At baseline, 97.6% of participants met the PTSS threshold, decreasing to 26.4% at three months and 17.6% at six months. This high baseline prevalence likely reflects the early post-injury assessment period and the use of a sensitive screening threshold. At three months, road traffic accidents (OR 3.71, 95% CI 2.60–6.85) and car accidents (OR 2.20, 95% CI 1.68–3.15) significantly increased PTSS risk compared to falls. Higher baseline anxiety (OR 0.53, 95% CI 0.33–0.86) and kinesiophobia (OR 0.58, 95% CI 0.39–0.86) were associated with reduced PTSS odds. At six months, higher chronic pain–related disability independently predicted lower PTSS risk (OR 0.89, 95% CI 0.82–0.97). Injury severity differed significantly at six months (p = 0.022) but not at three months (p = 0.172). The six-month model demonstrated excellent discrimination (AUC = 0.91) and good calibration (p = 0.70).

## Conclusions

PTSS following musculoskeletal trauma are influenced by trauma mechanism, psychological factors, and injury severity over time, supporting early risk stratification and targeted psychological intervention.

## Introduction

Experiencing physical trauma can lead to Post-Traumatic Stress Symptoms (PTSS), which may manifest as intrusion, hyper-arousal, and avoidance [1]. These symptoms can exacerbate other symptoms following musculoskeletal injuries, resulting in increased levels of pain, disability, depression, and anxiety [2]. Individuals with PTSS are more likely to report heightened pain and experience greater severity in pain interference, catastrophising, kinesiophobia, and disability [3]. These outcomes can significantly affect daily life, including relationships, work performance, and overall health. Individuals with PTSS tend to be less engaged in family life, social/recreational activities, and work-related tasks [4,5]. Furthermore, delayed or poor recovery from musculoskeletal injuries is associated with the presence of PTSS [6,7]. For instance, PTSS following a road traffic accident (RTA) predicts lower post-injury quality of life [8]. Beyond the individual impact, the economic burden of PTSS is considerable. Although healthcare cost data specific to PTSS are limited, excess costs for Post-Traumatic Stress Disorder (PTSD) were estimated at $232.2 billion in the United States in 2018. In Germany, individuals with PTSD incur three times higher healthcare costs than those without PTSD over a five-year period [9]. In the UK, severe stress and PTSD claims in 2003/04 totalled £103 million [10]. In 2023/24, stress, depression, and anxiety accounted for 16.4 million days lost due to work-related ill health, averaging 21.1 days off per individual. Musculoskeletal disorders accounted for 7.8 million days lost, with an average of 14.3 days off [11].

A key challenge is identifying who is likely to develop PTSS following trauma. Influencing factors include trauma response, injury severity, recovery duration, social

support, and quality of life [12]. Injury severity and specific types (e.g., head and facial injuries) are predictors of more severe PTSS [13]. Other risk factors include gender, age, chronic illnesses, injury cause, coping styles, pain severity, cognitive function at discharge, and employment status [14–16]. Timing of assessment is also key; PTSD symptoms can remain elevated for up to six months, particularly after traffic-related injuries [17]. This study aims to determine predictors of PTSS at three and six months post-musculoskeletal trauma and hospitalisation of the injury (or injuries) by examining socio-demographic factors, pre-existing mental health conditions, disability, social support, coping strategies, and injury severity. Understanding these predictors can guide tailored interventions to reduce the impact of PTSS, ultimately improving recovery and quality of life.

## Materials and methods

### Study design

The present study is a secondary analysis of data from a cohort study [18]; this study utilised anonymised patient-reported outcome measures (PROMs) and demographic data from this study. Ethical approval for the original study was obtained from the relevant ethics committee (Approval number: 17/WA/0421). Written informed consent was obtained from all participants prior to their inclusion in the study. The present study adheres to the Strengthening the Reporting of Observational Studies in Epidemiology (STROBE) reporting guideline [19] (S1 file).

### Data source

A description of the data collection is provided in detail elsewhere [18]. In brief, data were analysed from consecutive patients with acute musculoskeletal trauma aged 16 years and older who were emergency admissions to a major trauma centre in the UK. The hospital admission register was used to identify all consecutive eligible patients between December 2018 and March 2020.

### Study Population and Analytical Procedures

Full details of the study population have been described elsewhere [18]. Briefly, the study included patients who were admitted to a trauma centre within the previous 14 days due to acute musculoskeletal trauma. Patients were also required to be able to understand and use written and spoken English, as well as have the mental capacity to provide informed consent (e.g., no confusion, delirium, severe cognitive impairment, or severe mental illness, defined by a score of ≤6 on the Abbreviated Mental Test [AMT-10]) [20]. Patients were excluded if they had acute intracranial bleeding and a Glasgow Coma Scale score of 14 or less, brain or central nervous system injury, long-term neurocognitive disorders (e.g., brain tumour, multiple sclerosis, Alzheimer's disease, or Parkinson's disease), ongoing rheumatological condition, prolonged corticosteroid use, comorbid cancer, or terminal illness with short life expectancy. These exclusion criteria were applied to ensure a relatively homogeneous sample and reduce confounding factors; however, they may limit the generalisability of the findings to broader trauma populations, particularly those with neurological injuries or cognitive impairment.

### Post-Traumatic Stress Symptoms definition

PTSS was measured at baseline, three, six, and 12 months using the Revised Impact of Event Scale (IES-R), a 22-item self-report scale designed to measure subjective distress caused by traumatic events [21]. Respondents identify a specific stressful life event (in this case, the traumatic event) and rate how much each listed "difficulty" has distressed them in the past 7 days. The IES-R uses a 5-point scale from 0 ("not at all") to 4 ("extremely"), producing a total score ranging from 0 to 88. Subscales for intrusion, avoidance, and hyperarousal are also calculated [22]. The IES-R has demonstrated high validity in diverse populations, including those with musculoskeletal injuries [23]. A cut-off score of 22 was used to categorise individuals at each time point as having or not having PTSS [24]. This cut-off was chosen based on Beck et al. [24],

as our sample (acute musculoskeletal trauma) likely represents individuals at early or subclinical stages of PTSD. A score of 22 or above has been used in other studies to flag elevated PTSS risk, even if it does not indicate full PTSD diagnosis. Since participants were evaluated in the acute post-injury phase (within 2 weeks), many may present transient or emerging symptoms. While a cut-off of 33 is typically used to suggest probable PTSD, the lower threshold of 22 offers greater sensitivity to identify early symptoms. This is especially appropriate for screening purposes in the acute phase post-trauma, where it is important to avoid missing individuals at risk. The aim was to identify participants for further monitoring rather than to make definitive diagnosis of PTSD. However, the use of a lower cut-off (IES-R ≥ 22) in the acute post-injury phase may result in a high prevalence of PTSS, reflecting transient distress rather than clinically persistent symptoms. This may introduce a ceiling effect at baseline and should be considered when interpreting the distribution of PTSS and subsequent analyses.

## Predictor variables

Potential predictors measured at baseline, three months, and six months were analysed, as these time points provided sufficient information for meaningful interpretation. The 12-month data were excluded due to significant missing data, rendering multiple imputation unreliable for robust analysis. Clinical variables were measured consistently at three and six months. A wide range of variables were included to capture a comprehensive set of potential predictors. Anxiety and depression were evaluated using the Hospital Anxiety and Depression Scale (HADS), assessed at baseline to reflect participants' psychological state immediately post-injury. S2 file provides a structured overview of all candidate predictor variables included in the study, grouped into clinical, trauma-related, and socio-demographic domains. Participants were not specifically asked to recall their pre-injury mental health status. Potential predictors were identified based on previous studies [25,26]. To ensure comprehensive coverage of biopsychosocial factors associated with PTSS, candidate predictors were grouped into three domains:

- Clinical variables included BMI, Glasgow Coma Scale (GCS), surgery status, SF-36 Physical and Mental Health scores, EQ-5D-5L, Tampa Scale of Kinesiophobia (TSK)-11, HADS (anxiety and depression), Brief Pain Inventory (BPI) pain intensity, IES-R score, Pain Self-Efficacy Questionnaire (PSEQ), and Chronic Pain Grade Scale (CPGS).

- Trauma-related variables included trauma type, injury location (upper/lower limb, back/neck, chest/abdomen, head/face), number of fractures, injury severity, number of surgeries, previous surgery, days in hospital, days since admission, and days since trauma.

- Socio-demographic variables included civilian/military status, alcohol consumption, smoking status, drug use, medical history, comorbidities, age, gender, education level, employment status, and ethnic group.

  These variables were selected to reflect the multifactorial nature of PTSS and to support robust multivariable modelling.

## Measurement tool

A comprehensive set of validated clinical, psychological, trauma-related, and socio-demographic variables were collected to support predictive modelling. Clinical variables included Body Mass Index (BMI), calculated from height and weight; Glasgow Coma Scale (GCS) scores to assess consciousness; and binary indicators for surgery and injury types. Patient-reported outcomes were captured using validated instruments: SF-36 (Physical Functioning and Mental Health subscales), EQ-5D-5L for health-related quality of life, TSK-11 for fear of movement, HADS for anxiety and depression, BPI for pain intensity, IES-R for post-traumatic stress, PSEQ for pain self-efficacy, and CPGS for pain complexity. Trauma-related variables included injury location (e.g., upper/lower limb, head/face), number of fractures, days in hospital, number of surgeries, and injury severity scores. Socio-demographic data captured via self-report or clinical records included age, gender, education, employment status, ethnic group, civilian/military status, and lifestyle factors such as

alcohol use, smoking, and drug use. Where applicable, variables were coded as continuous, binary, categorical, or count data, as detailed in the supplementary materials. A complete list of candidate predictors, including the coding mechanism and source, can be found in S3 file.

## Missing data and imputation process

When examining missing data in our dataset, we found that most missing values were associated with psychological health measures. These patterns appeared to correlate with observed factors such as age, injury severity, and demographics. For instance, younger patients were more likely to have missing data for the HADS and the SF-36 Mental Summary. This suggests that the missingness is related to observed variables, supporting the Missing at Random (MAR) assumption. To address this, we applied multiple imputation using available information to reduce potential bias. Multiple imputation was used for all variables with any level of missingness, including data reported in the Table of characteristics, such as self-reported questionnaires. This ensured the inclusion of participants with partial data, preserving statistical power and avoiding information loss. We created five imputed datasets (m = 5) using the Multivariate Imputation by Chained Equations technique [27], as increasing beyond five imputations provides minimal additional benefit [28]. This approach helped account for uncertainty in the missing data and enhanced the robustness of the analysis.

## Power calculation

To determine the adequacy of our sample size for detecting significant effects in our regression model, we conducted a priori power analysis using G*Power [29]. We specified a multiple regression framework with a significance level (α) set at 0.05, aiming for a statistical power of 0.80. A power analysis based on 16 predictor variables and a medium effect size ($f^2 = 0.15$) suggested that a sample size of around 109–120 participants would be needed to achieve adequate power. The sample included 125 participants, which indicates sufficient power (> 0.80) to detect at least medium effect sizes and provides adequate power for our regression analysis, allowing us to effectively evaluate the relationships among the multiple predictors and the outcome variable across the specified range of predictors.

## Statistical analysis

All statistical analyses were performed in STATA (version 13.1) [30]. Means and standard deviations (SD) were calculated for continuous variables, and medians and interquartile ranges (IQR) were calculated for frequencies for categorical variables. Categorical variables such as smoking status were recoded as yes/no and for normally distributed data, the mean and SD were reported, while median and IQR were used for non-normally distributed data. The alpha level was set at 5% for all statistical tests. The statistical analysis code used in this study is provided in Supplementary File 5 (S4 File).

## Regression analysis

. The strength of each potential predictor of PTSS was initially explored using linear regression as a preliminary screening step to examine crude associations with the binary PTSS outcome. This approach was used for exploratory purposes only and not for inference or model estimation, recognising that linear regression is not appropriate for modelling a dichotomous outcome. All subsequent analyses and reported results are based on binary logistic regression models, which are appropriate for modelling the presence or absence of PTSS. This preliminary analysis helped identify key predictors to explore further using logistic regression, aiming to develop a model for the binary outcome. For the three-month analysis, only baseline variables were used as candidate predictors, as demographic data were collected at baseline only. For the six-month analysis, both baseline and three-month clinical variables were considered. The inclusion of the 3-month PTSS data in the six-month model was based on the expectation that early symptoms provide insight into PTSS progression. Patterns at three months may indicate long-term outcomes, aiding in identifying individuals at higher risk. While not

intended for guiding early interventions, this approach supports understanding symptom evolution and strengthens the six-month predictive model. Regression coefficients and 95% confidence intervals (CIs) were reported. Candidate predictor variables were selected a priori based on prior literature on PTSS following trauma [31]. Univariate analyses were conducted to explore associations between each predictor and PTSS. These analyses were used as an initial screening step to identify candidate variables for multivariable modelling, using a significance threshold of $p < 0.05$; however, clinically relevant variables identified from prior literature were retained regardless of statistical significance. Variables demonstrating statistical significance at $p < 0.05$ were considered for inclusion in multivariable modelling. Clinically relevant confounders identified from previous research were retained irrespective of statistical significance [32]. Backward elimination was applied cautiously to reduce model complexity while minimising overfitting risk. Specifically, all selected candidate predictors were initially included in the multivariable model, and variables were sequentially removed based on the highest p-values, while monitoring changes in model fit (Akaike Information Criterion; AIC) and ensuring that clinically important variables were retained. A priori confounders relevant to PTSS were identified and retained regardless of significance. Backward elimination was used to select significant predictors, helping reduce overfitting while enhancing model accuracy [33]. This approach was used in an exploratory modelling framework given the relatively low events-per-variable ratio, and results should therefore be interpreted as hypothesis-generating rather than confirmatory. Given the relatively high number of candidate predictors in relation to the number of outcome events at follow-up timepoints, there is an increased risk of model overfitting, and therefore model estimates may be unstable and overly optimistic. To ensure validity, initial predictors were chosen based on theoretical frameworks and prior literature, reflecting underlying PTSS mechanisms. We conducted diagnostic checks, including multicollinearity assessment, to ensure model integrity. The final model's fit was assessed using the AIC and model performance was further evaluated using discrimination and calibration metric to assess model stability and reduce optimism in performance estimates. A binary logistic regression model was developed to examine the relationship between PTSS and relevant predictors. PTSS was measured at two time points: 3 and 6 months post-trauma with each time point analysed separately to explore temporal changes in predictive factors. Accordingly, results from the linear regression screening step were not interpreted and are not presented in the Results section.

## Number of outcome events and events-per-variable (EPV)

At 3 months, 33 participants met the PTSS threshold (events) out of 73 followed-up participants. The final multivariable model included 10 predictor variables, resulting in an EPV of 3.3 (33/10). At 6 months, 22 participants met the PTSS threshold (events) out of 82 followed-up participants. The final multivariable model included 12 predictor variables, resulting in an EPV of 1.8 (22/12). These values indicate a low events-per-variable ratio, particularly at 6 months, and suggest potential risk of overfitting.

## Managing multicollinearity in regression analysis

To assess multicollinearity in the regression model, the Variance Inflation Factor (VIF) was initially calculated from a linear regression model on the predictor variables. A VIF value exceeding 10 was considered indicative of high correlation with other independent variables, leading to the exclusion of these variables from further analysis to reduce multicollinearity and enhance the robustness of the results [34]. Significant predictor variables identified through univariate analysis were further evaluated for collinearity. Given the presence of multicollinearity among these variables, Principal Components Analysis (PCA) was employed as a strategy to mitigate these issues in the regression model. PCA replaces correlated variables with a smaller set of uncorrelated components [35]. Components with eigenvalues greater than 1 were retained to ensure that each selected component explained more variance than any individual original variable. Ultimately, eight principal components were retained, which accounted for 60% of the total variance in the dataset. PCA was implemented specifically to reduce redundancy among correlated psychological and pain-related variables and to enhance model stability. The retained components represent latent constructs rather than newly defined clinical variables. However, the use

of PCA reduces the interpretability of individual predictors, as the resulting components are linear combinations of multiple variables rather than directly observable clinical measures. To aid interpretation, the retained components were examined in relation to their dominant contributing variables (e.g., psychological distress, pain-related factors), allowing approximate mapping to clinically meaningful domains, although precise attribution to individual predictors is not possible. These components were subsequently entered into the multivariable regression models to improve interpretability and reduce instability of parameter estimates. This methodology was chosen due to the nature of the variables involved; some measured the same underlying construct such as pain severity using different testing methodologies. By incorporating PCA into the multivariate regression model, we aimed to provide a more interpretable and statistically robust model. By effectively mitigating multicollinearity and ensuring more reliable parameter estimates. In addition to addressing multicollinearity, PCA facilitated the identification of latent constructs underlying clusters of correlated variables, including measures of pain and psychological distress. This dimensional reduction enabled the modelling of broader domains of trauma response, thereby enhancing the interpretability and coherence of the regression framework used to examine risk factors for post-traumatic stress symptoms. While this approach improves statistical stability, it introduces a trade-off between reducing multicollinearity and maintaining clinical interpretability.

### Model Performance

The predictive models for PTSS at three and six months were evaluated by examining each model's discrimination, calibration, Area Under the Curve (AUC), and goodness of fit. Calibration was assessed using the Hosmer-Lemeshow goodness of fit test specifically for the logistic regression model. Discrimination for this model was evaluated using ROC curves, as indicated by the AUC. Discrimination measured the ability of the model to differentiate between individuals with PTSS and those without PTSS. Calibration evaluated how closely the predicted probabilities aligned with the observed outcomes, and AUC provided a comprehensive measure of the model's overall predictive performance. Goodness of fit examined how well the model fitted the observed data. Higher AUC values indicate better predictive models, with values greater than 0.75 suggesting useful predictions [36]. Given the low events-per-variable ratio and absence of external validation, performance metrics such as the AUC should be interpreted cautiously, as they may overestimate the true predictive ability of the mode.

### Injury severity

Patients were grouped into three categories of injury severity: mild (ISS < 9), moderate (ISS 9–15), and major (ISS > 15), based on the Injury Severity Score (ISS) as described by Bolorunduro et al. [37]. The Kruskal-Wallis test was used to compare changes in PTSS severity among the three injury severity groups (mild, moderate, and major) at three and six months post-injury. The focus was on evaluating the distribution and severity of PTSS scores across different injury severity levels to enhance understanding of the relationship between injury severity and PTSS. This analysis aimed to capture overall trends rather than strict classification, facilitating a more comprehensive insight into the association between injury severity and PTSS. Following this, Mann-Whitney U tests were conducted for pairwise comparisons to identify differences in PTSS severity between the injury severity groups at the specified time points.

### Results

A total of 125 participants with a mean (SD) age of 48.9 ± 18.7 years were included in this study. Follow-up response rates were 73 participants (58.9%) at three months, 82 participants (66.1%) at six months, and 44 participants (35.5%) at 12 months. In the current study, participant data were utilised only for the three and six-month timepoints since the significantly lower response rate at the 12-month mark (35.5%) raised concerns about data completeness and the potential for introducing bias and uncertainty associated with imputation in a context where the dropout rate is high. **Table 1** presents the baseline characteristics of the 125 participants.

**Table 1. Participant demographics and injury details.**

| Variables | n (%) | Mean (SD) or Median [IQR] |
|---|---|---|
| **Days since admission** | 125 | 5.7 (3.1) |
| **Days since trauma** | 125 | 6.2 (3.3) |
| **BMI (kg/m²)** | 125 | 27.8 (6.3) |
| **GCS** | 125 | 15 [15] |
| **Circumstance of injury**<br>Military<br>Civilian | 12 (9.6)<br>113 (90.4) | |
| **Trauma description** | | |
| Blunt trauma | 18 (14.4) | |
| Car accident | 8 (6.4) | |
| Falls | 61 (48.8) | |
| Gunshot wound. | 1 (0.8) | |
| Head on collision<br>Motorbike accident | 1 (0.8)<br>9 (7.2) | |
| Multiple stab wounds (individuals sustaining more than one stab wound) | 2 (1.6) | |
| Road traffic accident | 18 (14.4) | |
| Skiing accident | 5 (4.0) | |
| Stabbing injuries (refers to any injuries resulting from a stabbing incident) | 2 (1.6) | |
| **Upper injury** | | |
| No | 100 (80) | |
| Yes | 25 (20) | |
| **Lower limb injury**<br>No<br>Yes | 12 (9.6)<br>113 (90.4) | |
| **Back and neck injury** | | |
| No | 84 (67.2) | |
| Yes | 41 (32.4) | |
| **Chest abdominal injury** | | |
| No | 105 (84.0) | |
| Yes | 20 (16.0) | |
| **Head and face injury.**<br>No<br>Yes | 116 (92.8)<br>9 (7.2) | |
| **Number of fractures** | | 1.9 (1.2) |
| 1 | 65 (52) | |
| 2 | 23 (18.4) | |
| 3 | 12 (9.6) | |
| 4 | 7 (5.6) | |
| 5 | 8 (6.4) | |
| Missing | 10 (8.0) | |
| **Surgery** | | |
| No | 15 (12.0) | |
| Yes | 110 (88.0) | |

*(Continued)*

| Variables | n (%) | Mean (SD) or Median [IQR] |
|---|---|---|
| **Number of surgeries** | | |
| 0 | 16 (12.8) | |
| 1 | 61 (48.8) | |
| 2 | 24 (19.2) | |
| 3 | 12 (9.6) | |
| 4 | 6 (4.8) | |
| 5 | 6 (4.8) | |
| **Alcohol preinjury** | | |
| No | 79 (63.2) | |
| Yes | 46 (36.8) | |
| **Smoking** | | |
| No | 92 (73.6) | |
| Yes | 21 (16.8) | |
| Missing | 12 (9.6) | |
| **Drug use.** | | |
| No | 121 (96.8) | |
| Yes | 4 (3.2) | |
| **Medical history** | | |
| No | 64 (51.2) | |
| Yes | 61 (48.8) | |
| **Presence of comorbidities** | | |
| No | 73 (58.4) | |
| Yes | 52 (41.6) | |
| **Injury severity** | | |
| Major | 22 (17.6) | |
| Mild | 54 (43.2) | |
| Moderate | 49 (39.2) | |
| **Days in hospital** | 125 | 13 [9, 20.5] |
| **Age (years)** | 125 | 48.9 (18.7) |
| **Sex** | | |
| Female | 44 (35.2) | |
| Male | 81 (64.8) | |
| **Formal education (years)** | | |
| ≤16 | 49 (39.2) | |
| 17-19 | 40 (32) | |
| ≥20 | 30 (24.0) | |
| Full time education | 6 (4.8) | |
| **Working** | | |
| No | 44 (35.2) | |
| Yes | 81 (64.8) | |

*(Continued)*

Table 1. (Continued)

| Variables | n (%) | Mean (SD) or Median [IQR] |
|---|---|---|
| **Ethnic group** | | |
| British Asian | 10 (8.0) | |
| Chinese | 6 (4.8) | |
| Mixed race | 1 (0.8) | |
| Other ethnic group | 2 (1.6) | |
| White | 106 (84.8) | |
| **Previous injury** | | |
| No | 38 (30.4) | |
| Yes | 87 (69.6) | |
| **Clinical outcomes** | | |
| **HADS depression subscale (0–21)** | | 7.6 (4.3) |
| **HADS anxiety subscale (0–21)** | | 8.0 (4.9) |
| **SF-36 (physical)** | | 31.0 (7.6) |
| **SF-36 (mental)** | | 35.9 (13.4) |
| **EQ-5D-5L** | | 17 (3.8) |
| **BPI Pain Intensity subscale (average score 0–10)** | | 64.6 (21.1) |
| **Post-traumatic stress (IES score) (0–88)** | | 64.6 (21.1) |
| **PTSS (22-point cut-off)** | | |
| **Baseline** | 122 (97.6) | 44.4 (15.5) |
| **3 – months** | 33 (26.4) | 47.1 (15.4) |
| **6 – Months** | 22 (17.6) | 39.4 (15.8) |
| **TSK-11** | | 26.9 (7.2) |
| **PSEQ pain self-efficacy** | | 40.5 [28.5, 50.5] |
| **CPGS chronic pain** | | 63.3 [36.7, 83.5] |

**Abbreviations:** BMI: Body Mass Index; GCS: Glasgow Coma Scale; SD: Standard Deviation; IQR: Interquartile Range; HADS: Hospital Anxiety and Depression Scale; SF-36: 36-Item Short Form Survey; EQ-5D-5L: Euro-Qol 5 Dimension 5 Level; BPI: Brief Pain Inventory; IES: Impact of Events Scale.; TSK; Tampa Scale of Kinesiophobia; PSEQ; Pain Self Efficacy Questionnaire; CPGS; Chronic Pain Grade Scale

### Factors Predicting PTSS Three Months Following Musculoskeletal Trauma

Individuals who were involved in a car accident were 2.20 (95% CI 1.68–3.15, p = 0.015) times more likely to have PTSS at three months compared to those who had a fall. This suggests that for every one individual with PTSS among those who experienced a fall, approximately 2.20 individuals will have PTSS among those involved in a car accident. Additionally, those involved in RTAs, which include all road traffic accidents such as those involving motorbikes, were 3.71 (95% CI 2.60–6.85, p = 0.010) times more likely to have PTSS at three months compared to those who had a fall, indicating that for every one individual with PTSS in the fall group, about 3.71 individuals can be expected to have PTSS among those involved in RTAs. Furthermore, both HADS anxiety scores and TSK-11 scores were significant predictors of PTSS, with each one-unit increase in HADS anxiety score being associated with a decrease in the odds of having PTSS (OR 0.53, 95% CI 0.33–0.86, p = 0.011). Similarly, each one-unit increase in TSK-11 score was also associated with a decrease in the odds of having PTSS (OR 0.58, 95% CI 0.39–0.86, p = 0.007). However, these inverse associations are counterintuitive and should be interpreted with caution, as they may reflect residual confounding, measurement timing effects, or statistical artefact rather than true underlying relationships. The results of the multivariate analysis showing the association of the candidate predictors with PTSS at three months are presented in **Table 2**.

**Table 2. Predictors associated with the development of PTSS at 3 months – multivariate analysis.**

| Candidate predictors | OR | 95% confidence interval (CI) of OR | *P*-value |
|---|---|---|---|
| **Age (years)** <20 | Reference | | |
| 39.9-59.9 | 0.12 | (0.01, 1.64) | 0.113 |
| ≥60 | 1.67 | (0.28, 9.92) | 0.570 |
| **Trauma description** Falls | Reference | | |
| Car accident | 2.20 | (1.68, 3.15) | **0.015\*** |
| Road traffic accident | 3.71 | (2.60, 6.85) | **0.010\*** |
| **Upper limb injury** No | Reference | | |
| Yes | 0.19 | (0.03, 1.32) | 0.093 |
| **Lower limb injury** No | Reference | | |
| Yes | 7.00 | (0.56, 87.4) | 0.131 |
| **Chest abdominal injury** No | Reference | | |
| Yes | 9.80 | (0.63, 15.1) | 0.103 |
| **Medical history** No | Reference | | |
| Yes | 1.38 | (0.25, 7.53) | 0.710 |
| **Injury severity** Minor | Reference | | |
| Major | 0.68 | (0.05, 7.97) | 0.764 |
| Moderate | 1.31 | (0.28, 5.97) | 0.723 |
| **HADS depression subscale (0–21)** | 1.01 | (0.64, 1.61) | 0.947 |
| **HADS anxiety subscale (0–21)** | 0.53 | (0.33, 0.86) | **0.011\*** |
| **SF-36 (physical)** | 0.81 | (0.63, 1.04) | 0.108 |
| **SF-36 (mental)** | 0.91 | (0.75, 1.10) | 0.341 |
| **EQ-5D-5L** | 0.54 | (0.23, 1.29) | 0.170 |
| **BPI Pain Intensity subscale (average score 0–10)** | 0.94 | (0.87, 1.01) | 0.124 |
| **TSK-11** | 0.58 | (0.39, 0.86) | **0.007\*** |
| **PSEQ** | 0.92 | (0.77, 1.10) | 0.407 |
| **CPGS disability** | 0.96 | (0.91, 1.01) | 0.174 |

Abbreviations: HADS: Hospital Anxiety and Depression Scale; SF-36: 36-Item Short Form Survey; EQ-5D-5L: Euro-Qol 5 Dimension 5 Level; BPI: Brief Pain Inventory; IES: Impact of Events Scale.; TSK; Tampa Scale of Kinesiophobia; PSEQ; Pain Self Efficacy Questionnaire; CPGS; Chronic Pain Grade Scale

## Factors Predicting PTSS Six Months Following Musculoskeletal Trauma

Multicollinearity detection using VIF revealed high collinearity between the CPGS and the BPI pain intensity subscale. Detailed results of the VIF analysis can be found in Supplementary file 5. The PCA results indicated that depression explains the most variance among the components (eigenvalue = 5.98). Anxiety also contributed significantly but to a lesser extent, with an eigenvalue of 1.08. The first two components, depression and anxiety, together accounted for 78% of the total variance. Detailed PCA results are available in Supplementary file 6. There was a statistically significant association between the CPGS disability score and PTSS, with an odds ratio (OR) of 0.89 (95% CI 0.82–0.97, p = 0.013). This indicates that for each one-unit increase in the CPGS disability score, the odds of developing PTSS decrease by approximately 11%. In other words, as the CPGS disability score increases, the likelihood of experiencing PTSS diminishes, highlighting the inverse relationship between disability and PTSS. The results of the multivariate logistic regression, showing the association of the candidate predictors with PTSS at six months, are displayed in **Table 3**.

**Table 3. Predictors associated with the development of PTSS at 6 months – multivariate analysis.**

| Candidate predictors | OR | 95% confidence interval (CI) of OR | P-value |
|---|---|---|---|
| **Gender** Female | Reference | | |
| Male | 1.06 | (0.34, 3.27) | 0.910 |
| **Age (years).** <20 | Reference | | |
| 39.9-59.9 | 1.39 | (0.34, 2.65) | 0.643 |
| ≥60 | 2.91 | (0.66, 3.72) | 0.156 |
| **BMI (kg/m²)** <20 | Reference | | |
| 20.9-24.9 | 1.40 | (0.06, 2.90) | 0.826 |
| 25.0-29.9 | 2.37 | (0.13, 3.43) | 0.558 |
| 30.0-34.9 | 2.66 | (0.12, 4.82) | 0.532 |
| 35.0-39.9 | 1.78 | (0.11, 3.83) | 0.432 |
| 40.0-44.9 | 2.03 | (0.18, 4.04) | 0.115 |
| ≥50 | 1.56 | (0.22, 2.04) | 0.198 |
| **Years since Left education** Full time education | Reference | | |
| 17-19 | 1.71 | (0.39, 7.42) | 0.471 |
| ≤16 | 2.66 | (0.61, 6.84) | 0.191 |
| ≤20 | 1.81 | (0.78, 5.21) | 0.231 |
| **Smoking status** Non-smoker | Reference | | |
| Smoker | 0.28 | (0.05, 1.39) | 0.278 |
| Ex-smoker | 3.37 | (0.37, 4.33) | 0.123 |
| **Circumstances** Military | Reference | | |
| Civilian | 0.28 | (0.03, 2.57) | 0.265 |
| **Trauma description** Falls | Reference | | |
| Car accident | 0.86 | (0.29, 2.58) | 0.151 |
| Gunshot wound | 2.02 | (0.86, 4.71) | 0.124 |
| Head on collision | 1.16 | (0.28, 2.59) | 0.100 |
| Motorbike accident | 2.05 | (0.88, 4.80) | 0.449 |
| Road traffic accident | 1.35 | (0.51, 3.35) | 0.503 |
| Skiing accident | 1.07 | (0.90, 3.58) | 0.656 |
| Stabbing injuries | 1.22 | (0.68, 2.30) | 0.116 |
| Blunt trauma | 1.97 | (0.85, 4.97) | 0.151 |
| **Days since admission** <5 | Reference | | |
| 5-9 | 1.99 | (0.23, 3.32) | 0.530 |
| 11-14 | 1.64 | (0.15, 2.34) | 0.641 |
| **Days since trauma** <5 | Reference | | |
| 5-9 | 0.57 | (0.06, 2.87) | 0.609 |
| 11-14 | 0.32 | (0.04, 1.89) | 0.521 |
| **Glasgow Coma Scale** <10 | Reference | | |
| 11-14 | 1.63 | (0.15, 2.84) | 0.679 |
| 15 | 1.52 | (0.11, 1.96) | 0.521 |
| **Upper limb injury** No | Reference | | |
| Yes | 0.58 | (0.07, 4.85) | 0.621 |
| **Lower limb injury** No | Reference | | |
| Yes | 2.91 | (0.30, 7.55) | 0.351 |
| **Back and neck injury** No | Reference | | |
| Yes | 2.36 | (0.59, 6.48) | 0.222 |
| **Chest abdominal injury** No | Reference | | |

*(Continued)*

**Table 3.** (Continued)

| Candidate predictors | OR | 95% confidence interval (CI) of OR | P-value |
|---|---|---|---|
| Yes | 0.89 | (0.13, 5.82) | 0.904 |
| **Head and face injury** No | Reference | | |
| Yes | 1.82 | (0.09, 4.98) | 0.694 |
| **Number of fractures** <3 | Reference | | |
| 3-6 | 0.37 | (0.08, 1.61) | 0.187 |
| Missing | 0.46 | (0.02, 2.11) | 0.598 |
| **Surgery** No | Reference | | |
| Yes | 0.75 | (0.50, 1.10) | 0.142 |
| **Number of surgeries** <3 | Reference | | |
| 3-6 | 2.38 | (0.64, 3.46) | 0.191 |
| **Alcohol** No | Yes | | |
| Yes | 1.34 | (0.39, 4.60) | 0.638 |
| **Drug use** No | Ref | | |
| Yes | 0.50 | (0.02, 3.46) | 0.631 |
| **Medical History** No | Reference | | |
| Yes | 0.55 | (0.43, 1.56) | 0.136 |
| **Comorbidities** No | Reference | | |
| Yes | 0.32 | (0.03, 3.17) | 0.335 |
| **Injury severity** Minor | Reference | | |
| Major | 0.35 | (0.07, 1.63) | 0.185 |
| Moderate | 1.92 | (0.51, 7.12) | 0.329 |
| **Days in hospital** <20 | Reference | | |
| 20.9-39.9 | 1.73 | (0.40, 7.40) | 0.459 |
| 40.9-59.9 | 1.15 | (0.09, 3.21) | 0.910 |
| **Working** No | Reference | | |
| Yes | 0.22 | (0.04, 1.14) | 0.732 |
| **Ethnic group** Other | Reference | | |
| Chinese | 0.90 | (0.65, 1.25) | 0.492 |
| British Asian | 1.15 | (0.85, 1.50) | 0.382 |
| White | 1.25 | (0.95, 1.65) | 0.161 |
| **Previous surgery** No | Reference | | |
| Yes | 0.80 | (0.19, 3.36) | 0.761 |
| **HADS depression subscale (0–21)** | 0.69 | (0.41, 1.17) | 0.174 |
| **HADS anxiety subscale (0–21)** | 0.64 | (0.36. 1.15) | 0.140 |
| **SF-36 (physical)** | 0.96 | (0.84, 1.26) | 0.631 |
| **SF-36 (mental)** | 1.02 | (0.85, 1.24) | 0.780 |
| **EQ-5D-5L** | 1.21 | (0.68, 2.15) | 0.502 |
| **BPI Pain Intensity subscale (average score 0–10)** | 0.98 | (0.87, 1.06) | 0.426 |
| **TSK-11** | 0.82 | (0.62, 1.08) | 0.165 |
| **Pain Self Efficacy Questionnaire (PSEQ)** | 0.51 | (0.03, 1.17) | 0.954 |
| **CPGS pain score** | 1.17 | (0.96, 1.43) | 0.105 |
| **CPGS disability score** | 0.89 | (0.82, 0.97) | **0.013*** |

**Abbreviation**: HADS; Hospital Anxiety and Depression Scale; SF; Short Form; BPI; Brief Pain Inventory; TSK; Tampa Scale of Kinesiophobia; PSEQ; Pain Self Efficacy Questionnaire; CPGS; Chronic Pain Grade Scale.

## Presence of PTSS Following Musculoskeletal Trauma: Model Performance

The model fit the data well, as shown by the Hosmer-Lemeshow test ($\chi^2$ (8) = 5.55, p = 0.70), and it demonstrated excellent discriminatory ability with an AUC of 0.91, however, this should be interpreted cautiously given the small number of outcome events, low events-per-variable ratio, and absence of external validation, all of which may lead to optimistic performance estimates. The 3-month model included 33 outcome events and 10 predictors (EPV = 3.3), while the 6-month model included 22 outcome events and 12 predictors (EPV = 1.8). These values indicate a relatively low EPV, particularly in the 6-month model, and suggest that model estimates should be interpreted with caution due to potential overfitting and instability.

## Relationship between injury severity and PTSS after musculoskeletal trauma

The results of the Kruskal-Wallis test at the three-month follow-up showed that the extent of PTSS is similar across all categories of injury severity (p > 0.05). However, the analysis of PTSS at the six-month follow-up presented a different outcome. Results from the Kruskal-Wallis test indicate statistically significant differences in PTSS scores across injury severity levels (p = 0.022). Specifically, pairwise comparisons revealed significant differences between the "Mild" and "Moderate" injury categories (p = 0.018) and between the "Mild" and "Major" injury categories (p = 0.027). In contrast the subsequent evaluation of PTSS intensity at the three-month follow-up using Mann-Whitney U tests shows no significant differences among the mild, moderate, and major injury severity categories, although the comparison between mild and moderate injuries approached significance (p = 0.075). In contrast, at the 6-month follow-up, significant differences were identified, with significant differences observed between the mild and moderate injury groups (p = 0.018) and the mild and major injury groups (p = 0.027). No significant difference was found between the moderate and major injury severity groups (p = 0.538). Supplementary files 7–10 present the results of the Kruskal-Wallis and Mann Whitney U tests, which analysed differences in PTSS scores across different levels of injury severity (mild, moderate, and major). The Mann-Whitney U and Kruskal-Wallis test results are summarised in **Table 4** and **Table 5**, respectively. Statistical comparisons of PTSS scores across injury severity categories are detailed at both the three- and six-month follow-up points.

## Discussion

This study investigated factors that predict the presence of PTSS after musculoskeletal trauma and hospitalisation and to the authors' knowledge, represents the first empirical study to explore the relationship between injury severity and PTSS severity. A major finding was that higher levels of anxiety and kinesiophobia decreased the odds of having PTSS. These

**Table 4. Mann-Whitney U Test Results for PTSS Scores by Injury Severity.**

| Comparison | Timepoint | p-value | Decision |
|---|---|---|---|
| Mild vs Moderate | 3 months | 0.075 | Not significant |
| Mild vs Major | 3 months | 0.356 | Not significant |
| Moderate vs Major | 3 months | 0.857 | Not significant |
| Mild vs Moderate | 6 months | 0.018 | Significant |
| Mild vs Major | 6 months | 0.027 | Significant |
| Moderate vs Major | 6 months | 0.538 | Not significant |

**Table 5. Kruskal-Wallis Test Results for PTSS Scores Across Injury Severity Categories.**

| Timepoint | $\chi^2$ (df = 2) | p-value | Decision |
|---|---|---|---|
| 3 months | 3.524 | 0.172 | No significant difference |
| 6 months | 7.595 | 0.022 | Significant difference |

findings should be interpreted cautiously. Given the observational design of this study, the associations do not indicate a protective effect but rather reflect statistical relationships observed within this cohort. Causal inferences cannot be made. Notably, the observed inverse associations between anxiety, kinesiophobia, and PTSS are counterintuitive and warrant particularly cautious interpretation, as they may be influenced by bias, measurement limitations, or model-related artefacts rather than reflecting true protective effects. Additionally, falls and RTAs were associated with increased odds of having PTSS, suggesting that specific trauma mechanisms may play a significant role in psychological outcomes. Furthermore, higher disability scores at baseline reduced the odds of having PTSS at six months, indicating that the functional status of patients may influence their psychological recovery. These novel findings can assist in decision-making when tailoring rehabilitation programmes for patients who have suffered acute musculoskeletal trauma. However, these findings must be interpreted in the context of potential model overfitting due to the relatively large number of predictors considered in relation to the number of outcome events, which may inflate the apparent strength and stability of the identified associations.

## Falls and Road Traffic Accidents

Falls and RTAs emerged as significant predictors of the presence of PTSS, consistent with previous research in trauma populations [38,39]. These studies [38,39] have established a strong relationship between RTAs and increased risk of PTSS, reflecting shared factors like anxiety related to life-threatening experiences and injury severity. These findings may be attributed to common factors, such as the anxiety individuals experience in response to life-threatening situations and the severity of their injuries, both of which can increase the likelihood of developing PTSS. Rumination about the accident has been identified as one of the strongest predictors of subsequent PTSD, which may indicate that persistent, intrusive thoughts about the traumatic event exacerbate distress and contribute to the chronicity of PTSS [40]. Moreover, psychological distress following traumatic injuries often manifests as negative alterations in mood and cognition, along with hyperarousal, thus causing higher levels of distress in patients [41]. This interplay of rumination and emotional response highlights the complex mechanisms through which falls and RTAs can lead to the development of PTSS, highlighting the need for therapeutic strategies focused on managing these psychological factors in patients recovering from such trauma.

## Car Accidents

Individuals involved in car accidents were more than twice as likely to develop PTSS compared to those who experienced a fall. This highlights the unique psychological impact of road traffic accidents, which often involve a sudden impact, a perceived threat to life, and a loss of control.

Blanchard et al. [38] reported that 39% of motor vehicle accident survivors met criteria for PTSD one-month post-injury, with symptoms persisting in many cases. Ehlers et al. [42] found that cognitive responses, such as rumination and negative appraisals were predictive of PTSD following car accidents, independent of injury severity. Compared to falls, which may be perceived as less threatening and more familiar, car accidents often affect younger individuals and carry broader social and functional consequences, including disruption to work, mobility, and legal stressors [43]. Given the elevated risk, individuals recovering from car accidents should be offered early psychological screening and access to trauma-informed interventions. Evidence-based approaches such as cognitive behavioural therapy (CBT), psychoeducation, and structured follow-up can help reduce symptom severity and support recovery.

## Anxiety, Fear of Movement, and Disability

The findings that anxiety and fear of movement are factors which reduce the likelihood of having PTSS diverge from the prevailing academic discourse [44]and should be interpreted cautiously as hypothesis-generating associations rather than evidence of a protective effect. Whilst the extant literature has typically associated these factors with increased psychological distress and PTSS risk [45,46], the current findings may reflect study-specific associations rather than true inverse relationships and should not be interpreted as contradicting established evidence... Rather than challenging

existing paradigms, these findings are more appropriately interpreted as exploratory signals that may be influenced by residual confounding, measurement timing, or statistical artefact in a small sample. Although one possible interpretation is that individuals with higher anxiety and kinesiophobia may engage in adaptive coping strategies [47], this explanation remains speculative and cannot be confirmed within the constraints of the present observational design. Furthermore, any apparent inverse relationship should be interpreted cautiously, as it may arise from residual confounding, reverse causality, measurement timing, or statistical artefact rather than a true underlying mechanism. Contrary to other studies [48,49], which indicate that higher perceived disability is linked to greater psychological distress, the observed inverse association in this study is likely influenced by sample-specific effects and should be interpreted cautiously as a non-confirmatory finding. While it is possible that individuals who perceive their disability as manageable may exhibit greater resilience [50,51], this interpretation is speculative and alternative explanations such as reverse causality or unmeasured confounding cannot be excluded. These findings should not be used to guide intervention development directly but rather highlight the need for further research to clarify the directionality and mechanisms underlying these associations. Although initially unexpected, the inverse association between early disability and anxiety scores and PTSS at follow-up may reflect statistical artefacts, measurement timing effects, or residual confounding rather than true underlying dissociative processes. Previous research, including a meta-analysis by Ozer et al. [31], identified peri-traumatic dissociation as a strong predictor of PTSD. Individuals who dissociate may underreport distress in the acute phase yet remain vulnerable to delayed symptom emergence. This could explain why higher self-reported disability and anxiety, potentially markers of more engaged emotional processing, were associated with reduced PTSS risk. These findings should not be interpreted as evidence that early distress is protective; rather, they highlight uncertainty in the observed associations and the need for cautious interpretation and further validation. These inverse associations require very cautious interpretation and should be considered preliminary and hypothesis-generating only. Several alternative explanations are likely, including measurement timing effects, residual confounding, statistical artefact arising from correlated psychological variables, and reverse causality. Although PCA was applied to reduce multicollinearity, it does not address underlying bias, residual confounding, or instability arising from small sample sizes and low events-per-variable ratios. Replication in larger, independent prospective cohorts with adequate power and external validation is essential before any causal or clinical interpretation of these relationships can be made.

## Injury Severity

In the present study, greater injury severity was associated with greater PTSS severity. The deterioration of psychological well-being in individuals suffering more severe injuries can often be exacerbated by intense pain and psychological stress [52]. This aligns with previous findings [53], which indicated that while patients with more severe injuries may generally be expected to experience greater psychological distress, the relationship between injury severity and PTSD can be complex. Specifically, Delahanty et al. [54] found that patients diagnosed with PTSD often had lower injury severity scores, suggesting that factors such as perceived life threat and individual psychological responses may play a more significant role in determining psychological outcomes than the severity of physical injuries alone. Supporting this complexity, Gabert-Quillen et al. [55] examined both subjective and objective injury severity ratings in trauma victims and found that subjective injury severity, rather than objective ratings, was a significant predictor of PTSS at six weeks and three months post-trauma. Their research also highlighted the moderating effect of peritraumatic factors (psychological responses occurring during and immediately after a traumatic event), particularly peritraumatic dissociation, which influenced the relationship between subjective injury severity and PTSS. Therefore, while greater injury severity can exacerbate pain and stress. At three months, PTSS was similar across injury categories, which suggests that factors like peritraumatic dissociation might affect how patients view their injuries. This was supported by the Kruskal-Wallis test, which showed no significant differences in PTSS scores across injury severity groups at three months (p = 0.172), and by Mann-Whitney U tests, which found no significant pairwise differences (mild vs moderate: p = 0.075; mild vs major: p = 0.356; moderate vs

major: p = 0.857). However, by six months, we found significant differences in PTSS scores, especially between mild and moderate injuries. This was reflected in the Kruskal-Wallis test (p = 0.022), with Mann-Whitney U tests confirming significant differences between mild and moderate (p = 0.018) and mild and major (p = 0.027) injury groups, while no significant difference was found between moderate and major injuries (p = 0.538). This highlights the importance of adapting psychological support as patients' needs change over time.

## Clinical implications

The findings highlight the importance of implementing screening methods to determine individuals at risk of developing psychological morbidity after experiencing musculoskeletal trauma. While similar approaches have been explored in previous research [56], this study highlights the need for ongoing validation and adaptation of these screening methods to effectively identify and support at-risk patients in clinical settings. This proactive approach can ensure early identification of patients who may benefit from tailored psychological interventions [57]. Evidence-based practices, including cognitive-behavioural therapy and graded exposure techniques, could be integrated into these strategies to further enhance effectiveness and support holistic patient care [58]. The findings suggest the necessity for clinicians to prioritise early psychological interventions based on injury severity, which may significantly improve recovery trajectories. These findings support the integration of structured psychological screening within musculoskeletal trauma care pathways, particularly within the first 2–4 weeks post-injury. Validated tools such as the IES-R and HADS may facilitate early identification of individuals at elevated risk. Early referral to rehabilitation strategies, including CBT, psychoeducation, and graded exposure interventions, may help mitigate longer-term psychological morbidity and improve functional recovery.

## Strengths and limitations

The longitudinal design of this study is a strength, which allows for the examination of PTSS progression over time and thus allowing examination of temporal associations, though not causal inference. Another key strength of this study lies in its rigorous statistical approach, utilising multiple imputation for missing data, multivariate regression analysis with backward elimination to enhance predictive accuracy, and the application of VIF and PCA to effectively manage multicollinearity, ensuring robust model performance. By using PCA, we were able to move beyond individual symptom scores and instead capture underlying dimensions, such as emotional distress and functional impairment, that may better reflect the complexity of trauma recovery. This dimensional approach aligns with recent calls for more holistic modelling in trauma research [59]. However, limitations include the reduction in follow-up data due to participant dropout, which resulted in relatively small sample sizes at the follow-ups. Also, the modelling approach, including univariate screening and backward elimination, may introduce bias by inflating the apparent importance of selected predictors and excluding potentially relevant variables. Additionally, the use of data-driven variable selection methods, such as univariate screening and backward elimination, may increase the risk of model instability and overfitting, particularly in the context of a small sample size and low events-per-variable ratio. Furthermore, although strategies such as backward elimination, PCA, were applied to strengthen the regression models, the modest sample size relative to the number of candidate predictors increases the potential risk of overfitting. Furthermore, the use of PCA limits the clinical interpretability of the findings, as regression coefficients relate to composite components rather than individual, directly measurable variables. Although these components broadly represent underlying domains such as psychological distress and pain, the indirect nature of these constructs makes it difficult to translate findings into specific clinical targets. Future studies using larger samples may allow modelling approaches that retain individual predictors while adequately addressing multicollinearity, thereby improving clinical applicability. Therefore, the predictive findings should be interpreted cautiously and require validation in larger independent samples. Additionally, patients were excluded if they had acute intracranial bleeding and a Glasgow Coma Scale score of 14 or less, brain or central nervous system injury, long-term neurocognitive disorders (such as brain tumours, multiple

sclerosis, Alzheimer's disease, or Parkinson's disease), ongoing rheumatological conditions, prolonged corticosteroid use, comorbid cancer, or terminal illness with short life expectancy. While these criteria were necessary to ensure patient safety and data reliability, they may limit the generalisability of the findings to the wider trauma population, particularly individuals with neurological injuries or cognitive impairment who may be at elevated risk of PTSS. Furthermore, as this was a single-centre study conducted in a UK major trauma setting, the findings may not be fully generalisable to other healthcare systems, geographic regions, or trauma populations with different demographic and clinical characteristics. These factors may affect the generalisability of the results and warrants careful interpretation of the findings. The use of self-reported measures may also introduce bias, and the potential for recall bias during the follow-up periods must also be acknowledged. A key limitation of this study is the low EPV ratio in the multivariable models, particularly at six months (EPV = 1.8), which falls below commonly recommended thresholds for stable logistic regression modelling. This increases the risk of overfitting and reduces confidence in the stability of regression coefficients and predictive performance metrics. Consequently, findings from the predictive models, including the AUC estimates, should be interpreted as exploratory and require validation in larger independent cohorts. Furthermore, model performance measures, including discrimination metrics such as AUC, are particularly prone to overestimation and may not generalise to external populations. Another limitation is the extremely high prevalence of PTSS at baseline (97.6%), which is likely attributable to the early timing of assessment and the use of a sensitive IES-R cut-off (≥22). While appropriate for screening purposes, this threshold may capture transient post-traumatic distress rather than persistent or clinically significant symptoms. This may have introduced a ceiling effect, limiting variability in the outcome and potentially affecting the model's ability to discriminate between individuals at higher versus lower risk of PTSS. Consequently, the interpretation of predictor effects, particularly those suggesting inverse associations, should be approached with caution

## Conclusion

PTSS is a significant concern following musculoskeletal trauma and understanding the risk factors that contribute to the presence of these symptoms is crucial for improving post-injury support and interventions. This study demonstrates that the presence of PTSS following musculoskeletal trauma is multifaceted, influenced by the type of injury, timing of assessment, and individual variations in disability, anxiety, and fear of movement. The findings indicate that traumatic events such as falls and RTAs present a higher risk for PTSS at a three-month follow-up. Moreover, the interaction between disability levels and the timing of assessments highlights how prolonged functional impairments can affect psychological outcomes. Overall, these results show the importance of considering both the nature of the injury and the psychological factors at play over time when managing the recovery process for individuals experiencing PTSS after musculoskeletal trauma. By identifying factors such as falls and RTAs that elevate risk, alongside further understanding of the influence of anxiety, kinesiophobia, and perceived disability, we advocate for a personalised and comprehensive approach to patient care, one that is tailored to the individual needs and unique circumstances of each patient, ultimately enhancing treatment outcomes. Importantly, the observed inverse associations between anxiety, kinesiophobia, disability, and PTSS should be interpreted strictly as exploratory findings. Given the observational design, potential for overfitting, limited sample size, and low events-per-variable ratio, these associations are highly susceptible to residual confounding, measurement timing effects, reverse causality, and statistical artefact. They should not be interpreted as protective effects or used to inform clinical decision-making. Further replication in larger, independent cohorts with robust external validation is required before any firm conclusions can be drawn. Future studies should examine PTSS progression beyond the six-month mark to provide insight into the long-term psychological impacts of musculoskeletal trauma. Furthermore, future research should explore the underlying reasons why anxiety and fear of movement may reduce the likelihood of developing PTSS. Also, caution is warranted when generalising these findings beyond similar clinical settings, and further multi-centre studies are needed to confirm external validity.

## Supporting information

**S1 File. STROBE Reporting Guideline.**
(DOCX)

**S2 File. Candidate Predictors.**
(DOCX)

**S3 File. Predictor Variables and Their Descriptions.**
(DOCX)

**S4 File. STATA Codes.**
(XLSX)

**S5 File. VIF 6 months.**
(DOCX)

**S6 File. PCA 6 Months.**
(DOCX)

**S7 File. Kruskal Wallis and Mann Whitney tests.**
(DOCX)

**S8 File. PTSS_3_MONTHS.**
(XLSX)

**S9 File. PTSS_6_MONTHS.**
(XLSX)

## Acknowledgments

The authors would like to acknowledge those that were investigators of the original study: David Evans, Alison Rushton, Nicola Middlebrook, Jon Bishop, and Jaimin Patel.

## Author contributions

**Conceptualization:** Ferozkhan Jadhakhan.

**Data curation:** Ferozkhan Jadhakhan.

**Formal analysis:** Ferozkhan Jadhakhan.

**Investigation:** Ferozkhan Jadhakhan.

**Methodology:** Ferozkhan Jadhakhan.

**Software:** Ferozkhan Jadhakhan.

**Supervision:** Deborah Falla, Alexander Dallaway.

**Writing – original draft:** Ferozkhan Jadhakhan.

**Writing – review & editing:** Ferozkhan Jadhakhan, Deborah Falla, Alexander Dallaway.

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
