## [Decision Letter · Decision Letter 0]

25 Feb 2026

PONE-D-25-65953Predictors of Post-Traumatic Stress Symptoms After Musculoskeletal Trauma

Dear Dr. Jadhakhan,

Thank you for submitting your manuscript to PLOS ONE. After careful consideration, we feel that it has merit but requires a minor revision to fully meet PLOS ONE’s publication criteria as it currently stands. Therefore, we invite you to submit a revised version of the manuscript that addresses the points raised during the review process. Submit the track-change version and the revised version with a point-to-point response addressing all peer reviewers' comments.

Please submit your revised manuscript by Apr 11 2026 11:59PM If you will need more time than this to complete your revisions, please reply to this message or contact the journal office at plosone@plos.org. Please include the following items when submitting your revised manuscript:

We look forward to receiving your revised manuscript.

Kind regards,

Mohammad Sidiq, PhD Pain Sciences Physiotherapy

Academic Editor

PLOS One

Journal Requirements:

2. We note that your Data Availability Statement is currently as follows: All relevant data are within the manuscript and in Supporting Information files.

Reviewers' comments:

Reviewer's Responses to Questions

**Comments to the Author**

1. Is the manuscript technically sound, and do the data support the conclusions?

Reviewer #1: Yes

Reviewer #2: Yes

2. Has the statistical analysis been performed appropriately and rigorously? 

Reviewer #1: Yes

Reviewer #2: Yes

3. Have the authors made all data underlying the findings in their manuscript fully available?

Reviewer #1: Yes

Reviewer #2: Yes

4. Is the manuscript presented in an intelligible fashion and written in standard English?

Reviewer #1: Yes

Reviewer #2: Yes

5. Review Comments to the Author

Reviewer #1: 1.Overall, the manuscript is scientifically sound and designed appropriately. The design of study, measurement tools and analytical outline are supporting the aims. It clearly defines the outcomes. However, there is minor suggestion in conclusion especially for inverse associations (e.g., anxiety, disability), should be interpreted more cautiously and framed as associative rather than protective effects.

2.Clarification of the use of PCA within the regression models and acknowledgment of potential overfitting due to sample size would further strengthen the analysis.

3.Specify which supplementary files contain the underlying datasets and whether analysis code is available would improve transparency and reproducibility.

4.Minor language editing is required and reduction of repetitions in the Methods and Results sections would improve clarity and conciseness.

Reviewer #2: The study is methodologically sound with appropriate design and validated outcome measures. The data generally support the conclusions drawn. The manuscript is alligned with STROBE guideline. However minor correction are needed such as:

1. Clinical implications could be expanded, particularly regarding screening and early intervention strategies.

2. Further explanation of the variable selection strategy would strengthen methodological clarity.

3. Minor grammatical and typographical corrections are recommended.

4. Clearification of the variable selection strategies and PCA interpretation will imrove tranperency.

5. The unexpected inverse association between anxiety, kinesiophobia and PTSS requires more cautious interpretation and consideration of alternative explanation.

Overall this study quite valuable and can be a good contribution to trauma rehabilitation research.

6. PLOS authors have the option to publish the peer review history of their article (what does this mean?). If published, this will include your full peer review and any attached files.

Reviewer #1: No

Reviewer #2: No

---

## [Author Response · Author response to Decision Letter 1]

23 Mar 2026

A detailed response letter has been included as part of this submission

---

## [Decision Letter · Decision Letter 1]

10 Apr 2026

PONE-D-25-65953R1Predictors of Post-Traumatic Stress Symptoms After Musculoskeletal TraumaPLOS One

Dear Dr. Jadhakhan,  Thank you for submitting your manuscript to PLOS ONE. After careful consideration, we feel that it has merit but does not fully meet PLOS ONE’s publication criteria as it currently stands. Therefore, we invite you to submit a revised version of the manuscript that addresses the points raised during the review process.  Please submit your revised manuscript by May 25 2026 11:59PM.If you will need more time than this to complete your revisions, please reply to this message or contact the journal office at plosone@plos.org.  Please include the following items when submitting your revised manuscript:

A letter that responds to each point raised by the academic editor and reviewer(s). You should upload this letter as a separate file labeled 'Response to Reviewers'.A marked-up copy of your manuscript that highlights changes made to the original version. You should upload this as a separate file labeled 'Revised Manuscript with Track Changes.'An unmarked version of your revised paper without tracked changes. You should upload this as a separate file labeled 'Manuscript.'

We look forward to receiving your revised manuscript.

Kind regards,

Mohammad Sidiq, PhD, Physiotherapy, FAIMER Fellow

Academic Editor

PLOS One

**Additional Editor Comments:**

Dear authors, I appreciate the revisions and changes you have made; however, there are still some minor comments to address based on the reviewer's feedback.

Reviewers' comments:

Reviewer's Responses to Questions

**Comments to the Author**

1. If the authors have adequately addressed your comments raised in a previous round of review and you feel that this manuscript is now acceptable for publication, you may indicate that here to bypass the “Comments to the Author” section, enter your conflict of interest statement in the “Confidential to Editor” section, and submit your "Accept" recommendation.

Reviewer #1: All comments have been addressed

Reviewer #2: All comments have been addressed

2. Is the manuscript technically sound, and do the data support the conclusions?

Reviewer #1: Partly

Reviewer #2: Yes

3. Has the statistical analysis been performed appropriately and rigorously? 

Reviewer #1: No

Reviewer #2: No

4. Have the authors made all data underlying the findings in their manuscript fully available?

The PLOS Data policy requires authors to make all data underlying the findings described in their manuscript fully available without restriction, with rare exception (please refer to the Data Availability Statement in the manuscript PDF file). The data should be provided as part of the manuscript or its supporting information or deposited to a public repository. For example, in addition to summary statistics, the data points behind means, medians, and variance measures should be available. If there are restrictions on publicly sharing data—e.g., participant privacy or use of data from a third party—those must be specified.

Reviewer #1: Yes

Reviewer #2: Yes

5. Is the manuscript presented in an intelligible fashion and written in standard English?

Reviewer #1: Yes

Reviewer #2: Yes

6. Review Comments to the Author

Please use the space provided to explain your answers to the questions above. You may also include additional comments for the author, including concerns about dual publication, research ethics, or publication ethics. (Please upload your review as an attachment if it exceeds 20,000 characters.)

Reviewer #1: 1. Comments to the Author

The revised manuscript has improved substantially in clarity, transparency, and methodological reporting. The authors have addressed several previous concerns, particularly regarding missing data handling, multicollinearity, and cautious interpretation of findings. The topic is clinically relevant, and the longitudinal design is appropriate.

However, several important issues remain that need to be addressed before the manuscript can be considered fully suitable for publication.

Major comments:

1. Risk of overfitting and model stability

The sample size is relatively small, particularly at follow-up (n≈73 at 3 months; n≈82 at 6 months), in relation to the large number of candidate predictors included in the regression models. This raises concerns about overfitting and model instability. The authors should:

Report the number of outcome events used in each model

Provide an events-per-variable (EPV) estimate

Temper claims regarding model performance (e.g., AUC = 0.91), as these may be optimistic without external validation

2. Interpretation of inverse associations (anxiety, kinesiophobia, disability)

The finding that higher anxiety, kinesiophobia, and disability are associated with reduced odds of PTSS contradicts the broader literature. While the authors acknowledge this and provide possible explanations, these remain speculative. The interpretation should be further softened and clearly framed as hypothesis-generating rather than explanatory. Alternative explanations such as residual confounding, measurement timing, or statistical artifacts should be emphasized more strongly.

3. Use of linear regression for a binary outcome

The manuscript states that linear regression was used with a binary PTSS outcome during initial analysis. This is not appropriate and should be corrected or clarified. If used only as a screening step, this should be explicitly justified.

4. Insufficient detail on cross-validation

The manuscript mentions cross-validation but does not describe the method (e.g., k-fold, split-sample, bootstrap). This needs clarification, or the statement should be removed.

5. Baseline PTSS prevalence (97.6%)

The extremely high baseline prevalence suggests a potential ceiling effect related to the chosen cut-off (IES-R ≥22) and timing of assessment. The authors should explicitly acknowledge how this may affect discrimination and interpretation of predictors.

Minor comments:

1. Clarify the type of informed consent obtained (e.g., written)

2. Improve formatting and clarity of tables (e.g., inconsistencies in Table 2)

Reviewer #2: Comments to the authors

This is a well-written and clinically relevant study examining predictors of post-traumatic stress symptoms following musculoskeletal trauma. The longitudinal design and use of validated outcome measures are strengths, and the manuscript demonstrates improved transparency compared to earlier versions.

However, there are several methodological and interpretative issues that should be addressed to strengthen the manuscript.

Major comments:

1. Predictive modeling limitations

The ratio of predictors to sample size appears high, particularly at follow-up timepoints, increasing the risk of overfitting. While the authors acknowledge this limitation, the strength of the reported model performance (e.g., high AUC) may be overstated. This should be more clearly contextualized, and the limitations expanded.

2. Use of PCA in regression modelling

While PCA is used to address multicollinearity, it reduces interpretability of the predictors. The authors should explicitly acknowledge this limitation and clarify how PCA-derived components relate to clinically meaningful constructs.

3. Unexpected direction of key associations

The inverse relationship between anxiety, kinesiophobia, disability, and PTSS is counterintuitive. Although alternative explanations are discussed, these remain speculative and should be framed more cautiously. The possibility of bias, measurement issues, or model artifacts should be emphasized.

4. Clarity of statistical methods

Greater clarity is needed regarding:

The role of univariate screening

The backward elimination process

The cross-validation approach

5. Generalizability

The exclusion criteria (e.g., neurological conditions, severe cognitive impairment) and single-center design may limit generalizability. This should be discussed more explicitly

Minor Comments:

1. Clarify the consent procedure in the ethics statement.

2. Consider simplifying large tables for readability.

7. PLOS authors have the option to publish the peer review history of their article (what does this mean?). If published, this will include your full peer review and any attached files.

If you choose "no," your identity will remain anonymous, but your review may still be made public.

Reviewer #1: No

Reviewer #2: No

You may also use PLOS’s free figure tool, NAAS, to help you prepare publication-quality figures: https://journals.plos.org/plosone/s/figures#loc-tools-for-figure-preparation.

---

## [Author Response · Author response to Decision Letter 2]

15 Apr 2026

Response to reviewer letter uploaded

---

## [Editor Report · Decision Letter 2]

19 Apr 2026

Predictors of Post-Traumatic Stress Symptoms After Musculoskeletal Trauma

PONE-D-25-65953R2

Dear Jadhkhan,

We’re pleased to inform you that your manuscript has been judged scientifically suitable for publication and will be formally accepted for publication once it meets all outstanding technical requirements.

Within one week, you’ll receive an e-mail detailing the required amendments. When these have been addressed, you’ll receive a formal acceptance letter, and your manuscript will be scheduled for publication.

Kind regards,

Mohammad Sidiq, PhD, Physiotherapy, FAIMER Fellow

Academic Editor

PLOS One

Additional Editor Comments (optional):

Thank you, authors, for carefully addressing our comments and suggestions. I feel now this is suitable at this stage for acceptance. I congratulate the authors for this work. Wish you all the luck.
---

## [Editor Report · Acceptance letter]

PONE-D-25-65953R2

PLOS One

Dear Dr. Jadhakhan,

I'm pleased to inform you that your manuscript has been deemed suitable for publication in PLOS One. Congratulations! Your manuscript is now being handed over to our production team.

Kind regards,

on behalf of

Dr. Mohammad Sidiq

Academic Editor

PLOS One